# GenDP: 3D Semantic Fields for Category-Level Generalizable Diffusion Policy

**Yixuan Wang[1], Guang Yin[2], Binghao Huang[1], Tarik Kelestemur[3], Jiuguang Wang[3], Yunzhu Li[1]**

[1]Columbia University, [2]University of Illinois, Urbana-Champaign, [3]Boston Dynamics AI Institute

**Abstract:** Diffusion-based policies have shown remarkable capability in executing complex robotic manipulation tasks but lack explicit characterization of geometry and semantics, which often limits their ability to generalize to unseen objects and layouts. To enhance the generalization capabilities of Diffusion Policy, we introduce a novel framework that incorporates explicit spatial and semantic information via 3D semantic fields. We generate 3D descriptor fields from multi-view RGBD observations with large foundational vision models, then compare these descriptor fields against reference descriptors to obtain semantic fields. The proposed method explicitly considers geometry and semantics, enabling strong generalization capabilities in tasks requiring category-level generalization, resolving geometric ambiguities, and attention to subtle geometric details. We evaluate our method across eight tasks involving articulated objects and instances with varying shapes and textures from multiple object categories. Our method demonstrates its effectiveness by increasing Diffusion Policy's average success rate on *unseen* instances from 20% to 93%. Additionally, we provide a detailed analysis and visualization to interpret the sources of performance gain and explain how our method can generalize to novel instances. Project page

**Keywords:** Semantic Fields, Category-Level Generalization, Imitation Learning, Diffusion Models

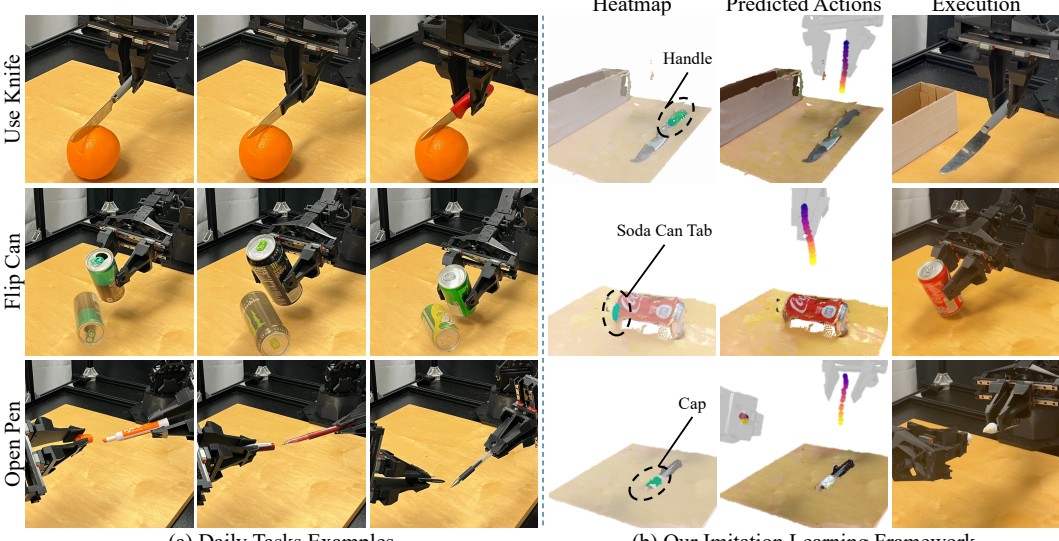

Figure 1: **Generalizable Diffusion Policy using 3D Semantic Fields.** Our approach introduces a diffusion policy capable of generalizing to new instances within a category by utilizing 3D semantic fields. These fields distinguish semantically meaningful parts of objects in 3D space, as illustrated in the heatmap example. Panel (a) on the left showcases daily task examples where semantic understanding is crucial, while panel (b) on the right demonstrates our method's ability to highlight semantically meaningful parts, such as a knife handle, and how our predicted policy accomplishes these tasks using the 3D semantic fields.

8th Conference on Robot Learning (CoRL 2024), Munich, Germany.

# 1 Introduction

Diffusion-based policies have recently shown promising results in imitation learning for real robot deployment in complex robotic manipulation tasks [1, 2], because they can express multimodal action distribution, output high-dimensional actions, and train stably. However, existing end-to-end diffusion policy frameworks do not generalize to novel instances due to their brittleness to environment variances, such as object appearances and background changes. Our work proposes a diffusion policy framework capable of category-level generalization and robust to environmental changes using 3D semantic fields.

Previous work has explored the use of point clouds as policy inputs to help generalization [2, 3], but the reliance on geometric information is insufficient for fine-grained scene understanding and generalization. For instance, as illustrated in Figure 1 (b), a marker's head and tail are geometrically ambiguous despite their functional and semantic differences. In this work, we augment geometric information with semantics to generalize to novel instances, resolve geometric ambiguity, and attend to subtle geometric details.

An ideal representation should not only extract geometric information from raw observation but also retain semantic information for better category-level generalization. In this work, we introduce a diffusion policy framework that uses a scene representation in the form of **3D semantic fields**. The 3D semantic fields are defined as the similarity fields, where the similarity score is higher for points closer to one object part, such as knife handles. Note that the 3D semantic fields have multiple channels, as shown in Figure 2, where the first channel corresponds to the shoe tongue, and the last one corresponds to the shoe cuff. Our framework consists of three main modules: a 3D descriptor fields encoder, a semantic fields constructor, and an action policy. The 3D descriptor encoder takes in multi-view RGBD observations and outputs a point cloud with high-dimensional descriptors using large foundational vision models like DINOv2 [4]. These descriptors are then fed into the semantic fields constructor and converted into low-dimensional semantic fields. Finally, the policy takes in the semantic fields along with the point cloud as inputs and predicts actions.

The proposed framework offers three benefits: (1) **Category-level generalization:** As semantic fields contain both 3D and semantic information, it guides our policy to focus on semantically meaningful parts essential for task completion, allowing generalization across instances within a category. (2) **Resolve geometric ambiguity:** Geometric information can be ambiguous. For example, the knife blade and knife handle are geometrically similar despite functional and semantical differences. Our semantic fields can localize space semantically close to parts important for task completion, such as knife blades, to disambiguate vague geometric information. (3) **Attention to subtle semantic details**: Semantic details might be lost due to real-world observation noise, such as toothbrush head and soda can tab. Some tasks are impossible to accomplish without sufficient details, such as spreading toothpaste on a toothbrush and flipping a lying soda can upwards, as shown in Figure 1. Because our semantic fields highlight semantically distinct regions, our method can pay attention to nuanced semantic details for task completion.

We systematically evaluate our method across eight tasks. Our task settings involve novel instances, ambiguous object geometry, and require attention to subtle semantic details. Compared to baseline methods, which frequently fail to generalize to novel instances due to a lack of both 3D and semantic information in their representation, our approach demonstrates significant advantages in category-level generalization. It also leads to substantial performance improvements in scenarios containing geometric ambiguity and subtle details. Through comprehensive experiments, we show an improvement in Diffusion Policy's average success rate on *unseen* instances from 20% to 93%. We also provide a detailed analysis of how well and why our method can generalize to novel instances.

In this work, our contributions are threefold: (1) We propose a diffusion policy framework that uses 3D semantic fields, which leverages visual foundation models to encode the environment's geometric and semantic information. (2) Our design allows category-level generalization, differentiating geometric ambiguity, and attending to geometric details. (3) We conduct comprehensive experiments and improve the original diffusion policy's average success rate on *unseen* instances from 20% to 93%. We also provide a detailed analysis of how well and why our method can generalize to novel instances.

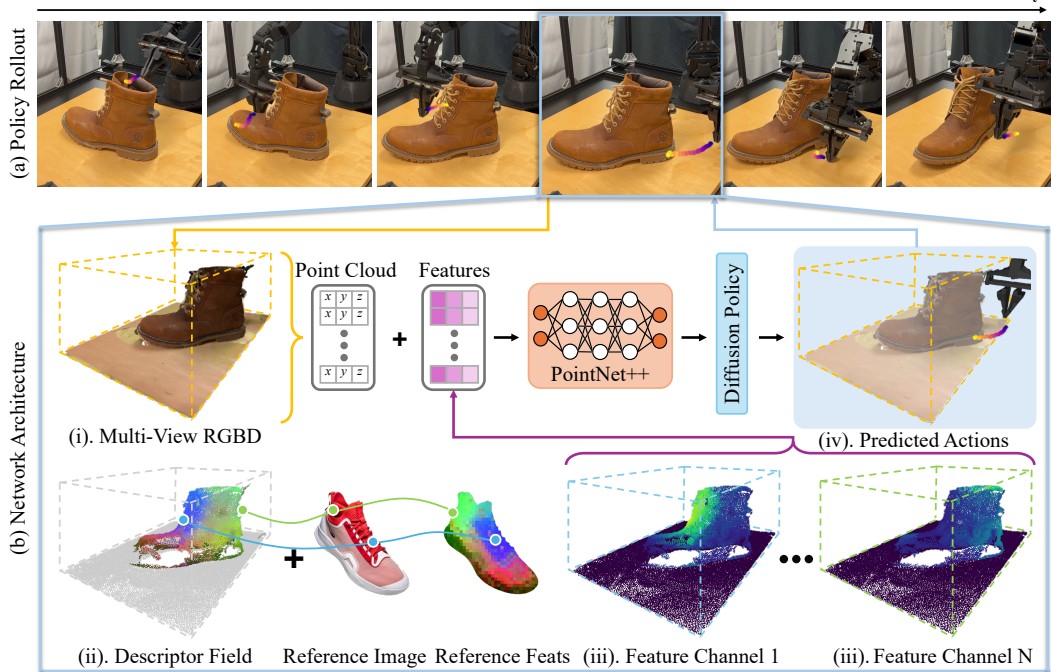

Figure 2: **Method Overview.** The top row (a) shows a sequence of real policy rollouts in the aligning shoe task. We first take in multi-view RGBD observations (i), then extract the 3D descriptor field, with each point possessing a corresponding high-dimensional descriptor (ii) [5]. We then select reference features from 2D reference images. By computing the cosine similarity between the descriptor field and 2D reference semantic features, we could obtain several semantic fields (iii). These semantic fields, concatenated with the point cloud, are then input into PointNet++ and the diffusion policy to output predicted actions (iv).

## 2 Related Works

**Diffusion Models for Robotics.** Beginning with random samples, diffusion models operate by iterative denoising to reveal the underlying distribution. They are widely used in robotics, including navigation [6], manipulation [1, 2, 7–14], control [15], and planning [16–18]. Among these works, Diffusion Policy and 3D Diffusion Policy are closely related to our work [1, 2].

Diffusion Policy produces a visuomotor policy under the action diffusion model, which can model the multi-modal distribution of demonstration actions in the real world. However, it relies on multi-view RGB observations, which are brittle to environment changes, including novel object instances, camera viewpoints, and background changes, and fail to generalize to novel instances. Instead of RGB observations, our work enables category-level generalization using 3D semantic fields.

3D Diffusion Policy approaches this problem using point clouds as inputs [2]. While this is proven to be sample-efficient, point clouds without semantic information can be insufficient for many manipulation tasks. In this work, we augment geometric information with semantic information using foundational vision models.

**Geometric Representation in Robotic Manipulation.** One typical geometric representation used in robotics is point clouds [3, 19–26], which enables the policy to focus on geometry and facilitates generalization across different environments. Yet, this often results in the loss of critical RGB data, limiting the understanding of an object's semantic properties. Unlike these approaches, our approach retains both semantic and geometric information, enhancing downstream decision-making.

Another line of works uses keypoints as the geometric representation in robotic manipulation [27–34]. They have shown impressive generalization capabilities and high efficiency due to their low dimension. However, the resulting sparsity loses geometric details and limits the range of tasks that can be effectively addressed. In contrast, our framework employs a representation with rich geometric information and enables a broader range of tasks.

**Semantic Representation in Robotic Manipulation.** Semantic reasoning is an ongoing field in robotics [35]. One common semantic representation learning frame is directly learning object-centric or scene-centric embeddings from RGB observations [36–44]. However, these representations are hard to generalize to new environments with different backgrounds and lighting conditions. In contrast, we use 3D semantic representation, which is generalizable across diverse environments and instances. Another class of semantic representation extracts functional or affordance information from observations [43, 45–52]. This line of work focuses on tasks that can be accomplished using motion primitives such as grasping, picking, and placing, while our diffusion-based policy is more flexible in action representation and tasks.

A recent line of research used neural implicit models such as Occupancy Networks [53] or NeRFs [54] to encode semantic features [5, 9, 25, 55–61]. GeFF is the closest one to our work. It operates by utilizing FeatureNeRF [62], which distills a NeRF model from foundational models. Although the distillation process allows for building NeRF upon sparse input views, it reduces the generalization capability. In contrast, we extract 3D descriptor fields without distillation, which inherits generalization capabilities from foundational vision models. In addition, GeFF uses predefined open-push-close action sequences to solve manipulation tasks. Our method differs by deploying an imitation learning policy that could accomplish more complicated tasks.

# 3 Method

## 3.1 Problem Statement

We define our system as a Markov Decision Process (MDP) consisting of state $s \in \mathcal{S}$ and action $a \in \mathcal{A}$. The system transition is defined through the dynamics model $s_{t+1} = f(s_t, a_t)$. We assume predicted actions follow the probability distribution $a_t \sim \mathcal{P}_{\text{pred}}(a|s_t) = \pi(s_t)$, where $\pi$ is the learned policy. The goal is to minimize the difference between $\mathcal{P}_{\text{pred}}(a|s_t)$ and ground truth action probability $\mathcal{P}_{\text{gt}}(a|s_t)$, given as a set of human demonstrations $D = \{\tau_0, \tau_1, ..., \tau_N\}$, where $\tau_i$ represents a trajectory comprising $\{s_0, a_0, s_1, ..., a_T\}$. Here, the state $s$ consists of a sequence of multi-view RGBD observations, and the action $a$ is a sequence of robot states.

## 3.2 3D Descriptor Fields

We use an off-the-shelf large foundational vision model, DINOv2 [4], to obtain semantic features from multi-view RGBD observations. Given its ability to extract consistent semantic features from the RGB images across context and instance variances, we selected it as the backbone network.

We provide pseudocode for building 3D descriptor fields in Algorithm 1. We denote single-view RGBD observation as $\boldsymbol{o}_i = (\mathcal{I}_i, \mathcal{R}_i)$, with $i \in \{1, 2, ..., N\}$ representing the camera index, consisting of an RGB image $\mathcal{I}_i \in \mathbb{R}^{H \times W \times 3}$ and a depth image $\mathcal{R}_i \in \mathbb{R}^{H \times W}$. We first use DINOv2 to extract dense 2D feature maps $\mathcal{W}_i$ corresponding to the RGB image $\mathcal{I}_i$ [4]. For an arbitrary 3D point $p$, we project it onto the image space to find its corresponding pixel location $u_i$ and the distance to camera $r_i$. We then interpolate to derive features $f_i$ from the feature map and the depth $r_i'$ from $\mathcal{R}_i$. The depth difference $\Delta r_i = r_i' - r_i$ reflects how distant $p$ is from the surface. When $p$ is closer to the surface in view $i$, greater weight is given to $f_i$. We fuse features from multiple viewpoints by applying a weighted sum, thus obtaining the descriptor $f$ corresponding to $p$. In practice, we follow the implementation details in D³Fields [5] to extract point cloud $\mathcal{P} \in \mathbb{R}^{K \times 3}$ and associated features $\mathcal{F} \in \mathbb{R}^{K \times F}$, where $K$ is the point cloud's size.

## 3.3 3D Sematic Fields

First, we obtain several 2D images containing the task-relevant object category and extract feature maps. Then we select features from each feature map and take the average to get a set of reference descriptors $\mathcal{F}_{\text{ref}} \in \mathbb{R}^{M \times F}$. Each descriptor represents a part of the object, such as the shoe head.

We denote the set of reference descriptors $\mathcal{F}_{\text{ref}}$ with $M$ descriptors and descriptor fields $\mathcal{F}$ with $K$ descriptors such that $\mathcal{F} = \{\mathcal{F}_i \in \mathbb{R}^F | i \in \{1, ..., K\}\}$ and $\mathcal{F}_{\text{ref}} = \{\mathcal{F}_{\text{ref},j} \in \mathbb{R}^F | j \in \{1, ..., M\}\}$. The semantic fields $\mathcal{C} \in \mathbb{R}^{K \times M}$ are defined as the similarity between the descriptor fields and reference descriptors:

$$\mathcal{C}_{ij} = \frac{\mathcal{F}_i \cdot \mathcal{F}_{\text{ref},j}}{||\mathcal{F}_i|| ||\mathcal{F}_{\text{ref},j}||}. \tag{1}$$

By computing the similarity scores, we convert high-dimensional descriptor fields into $M$-dimensional semantic fields, where $M$ is typically less than 5. We then concatenate semantic fields $\mathcal{C}$ and raw point cloud $\mathcal{P}$ together, which are then inputted into the policy.

## 3.4 Policy Learning

In this work, we model our policy as Denoising Diffusion Probabilistic Models (DDPMs), similar to Diffusion Policy [1, 63]. Instead of regressing the action directly, we train a noise predictor network

$$\widehat{\epsilon^k} = \epsilon_\theta(a^k, s, k), \quad (2)$$

**Algorithm 1** 3D Descriptor Fields Computation

1: Infer feature map $\mathcal{W}_i$ from single RGB image $\mathcal{I}_i$
2: **procedure** EVALUATE($p$)
3:     Project $p$ to camera $i$ and compute projected pixel $u_i$ and distance to camera $r_i$
4:     Obtain interpolated features $f_i = \mathcal{W}_i[u_i]$
5:     Obtain depth $r_i' = \mathcal{R}_i[u_i]$
6:     Compute depth difference $\Delta r_i = r_i' - r_i$
7:     Fuse features $\mathcal{F}_j = h(f_i, \Delta r_i), i \in \{1, 2, ..., N\}$

that takes in noisy actions $a^k$, current observations $s$, and denoising iterations $k$ and predicts the noise $\widehat{\epsilon^k}$. During training, we randomly choose a denoising step $k$ and sample noise $\epsilon^k$ added to the unmodified sample $a^0$. Our training loss is the difference between $\epsilon^k$ and predicted noise:

$$\mathcal{L} = \text{MSELoss}(\epsilon^k, \widehat{\epsilon^k}). \quad (3)$$

During the inference time, our policy starts from random actions $a^K$ and denoises for $K$ steps to obtain the final action predictions. At each step, the action is updated following

$$a^{k-1} = \alpha\big(a^k - \gamma\epsilon_\theta(a^k, s, k) + \mathcal{N}(0, \sigma^2 I)\big), \quad (4)$$

where $\alpha$, $\gamma$, and $\sigma$ are hyperparameters. In practice, we input 3D semantic fields into PointNet++ and obtain visual features. Then we feed visual features into diffusion policy, similar to [1].

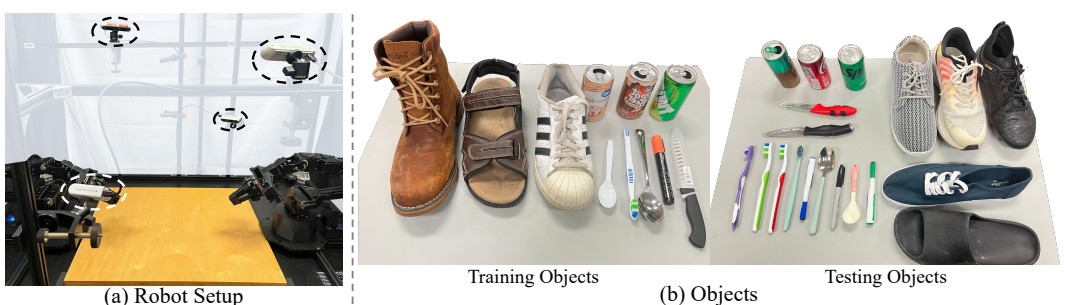

(a) Robot Setup        Training Objects     (b) Objects     Testing Objects

Figure 3: **Real Experiment Setup.** (a) We use four RealSense cameras to capture RGBD observations and ALOHA robots to execute policy. (b) We test on a diverse set of objects, including shoes, soda cans, marker pens, knives, spoons, toothbrushes, and toothpaste, with diverse geometry and appearance.

## 4 Experiments

In this section, we evaluate our method on eight diverse tasks. We aim to answer the following three questions through experiments. 1) How does the performance of our method compare to that of the state-of-the-art imitation learning methods? 2) How well can our method generalize to different configurations and instances? 3) What enabled our method to generalize to novel instances?

### 4.1 Setup

For experiments in simulation, we use SAPIEN [64] to build the environments and evaluate the policies. For the real-world experiments, we use the ALOHA robot and four RealSense cameras for real-world data collection and testing [65]. Figure 3 shows our real-world setup and objects used.

We evaluate our method on eight tasks, as shown in Figure 4. We test in the simulation on **Hang Mug** and **Insert Pencil** tasks. In the real world, we evaluate our method's ability from different perspectives, such as insignificant geometric details, category generalization, and geometric ambiguity. We collected 200 demonstration episodes for the **Hang Mug** task, 100 episodes for **Insert Pencil** task, and 60 episodes for all real-world tasks. More details are listed in the supplementary material. For each task, we train one policy to accomplish that single task.

We test our method on various object categories, including shoes, toothbrushes, soda cans, and knives, among others, which present significant challenges for category-level generalization due to

factors such as large geometric variances, ambiguous geometric information, and nuanced geometric details. More detailed descriptions of the tasks can be found in the supplementary material.

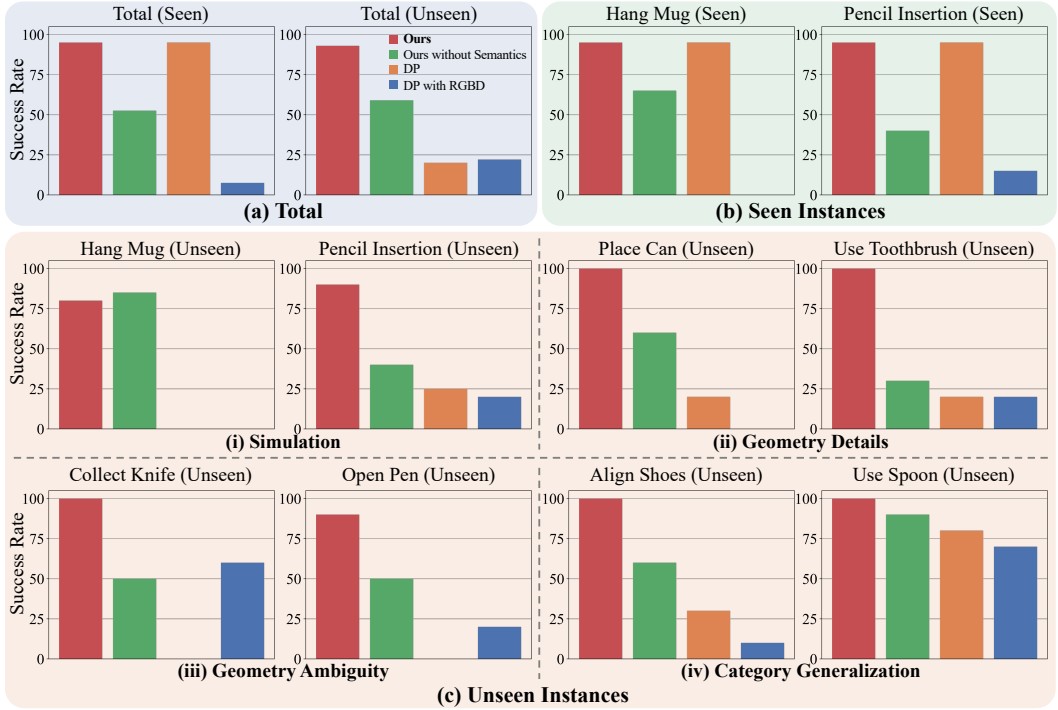

Figure 4: **Success Rate.** Our method was evaluated across eight tasks. (a) The aggregated quantitative results show that our method has similar performance as diffusion policy for seen instances, but our method outperforms all baselines on unseen instances. (b) For the seen instances, diffusion policy and our method have similar performances on two simulation tasks. (c) Diffusion policy performance degrades significantly on unseen instances. In addition, our method outperforms all other baselines, which underscores our policy's capability to attend to geometric details, distinguish geometric ambiguities, and generalize to novel instances.

## 4.2 Comparison with Baselines

We compare our method with three baselines: 1) **Ours without Semantics**: input raw point cloud without semantic fields, 2) vanilla **Diffusion Policy (DP)**, and 3) **DP with RGBD**: vanilla DP but with depth observations. For all baselines, we utilize four cameras to obtain observations and input them into the policy. We evaluate different policies by the success rate. The quantitative result is summarized in Figure 4 with detailed numbers in the supplementary material.

Our method shows similar performance with the original diffusion policy in the simulation on seen instances. However, when generalizing to unseen instances, diffusion policy performance degrades significantly, because it takes RGB images as input, which is brittle to environmental factors like object appearances and geometries. Therefore, the diffusion policy predicts undesirable actions and fails to accomplish the task. In contrast, our framework encodes explicit 3D and semantic information about the scene, which is robust to object appearance and geometry variances, which helps the policy to generalize to novel instances.

In addition, our method consistently outperforms ours *without* semantics, whether for seen instances or unseen instances. The benefits from 3D semantic fields are threefold. First, they help to attend to geometric details, such as the can tab, to place the can on the table upright, while ours without semantics cannot when solely relying on geometry information. Second, 3D semantic fields help to distinguish geometric ambiguity. For instance, when grasping the knife, it is hard to tell the knife directions from ambiguous geometry, which might lead to unsafe actions like grasping the knife blade. Our 3D semantic fields encode semantic information like knife handles to distinguish geometric ambiguities and show a higher success rate in Figure 4 (c)(iii). Third, Figure 4 (c)(iv) demonstrates our 3D semantic fields could help policy focus on semantically meaningful parts to complete tasks and generalize to novel instances.

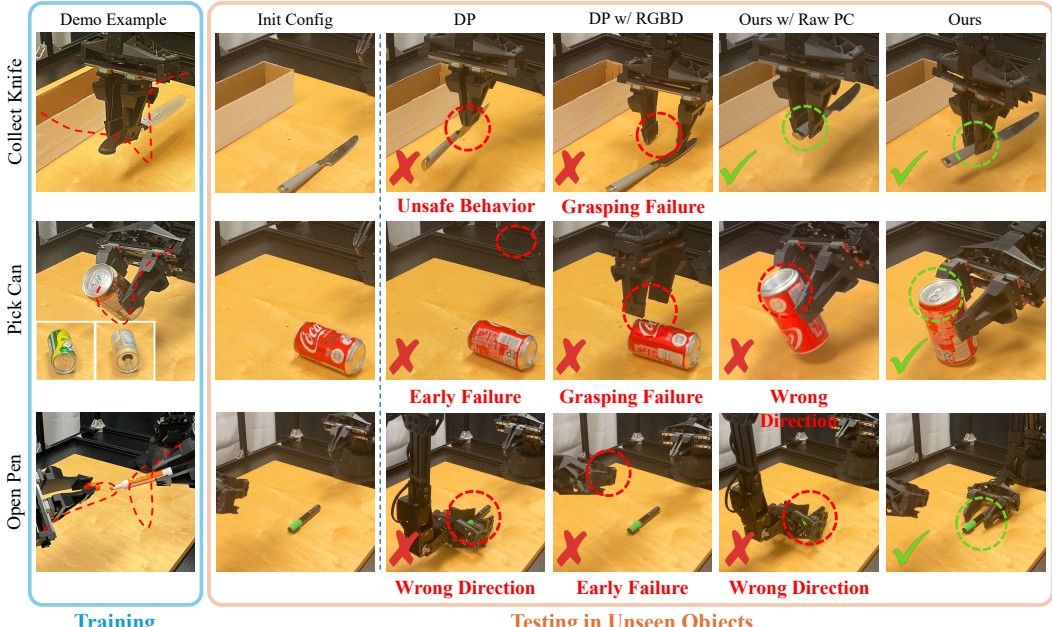

Figure 5: **Policy Rollout in Real World.** The figure illustrates the policy rollout results in the real world. On the left, the blue block displays the demonstration examples and corresponding training instances. On the right, the orange block presents policy rollout results. From left to right, they are, respectively, initial configurations, diffusion policy, diffusion policy with RGBD, ours *without* semantics, and our method. We summarize four common failure modes. Early failure and grasping failure could happen when the novel instance is presented. Diffusion policy may also lead to unsafe behavior when encountering novel instances. Ours *without* semantics might identify wrong directions due to geometric ambiguity and nuanced geometric details.

We found that Diffusion Policy with RGBD is not as effective as our method. Directly adding depth observation to the diffusion policy will make the input space even larger. Therefore, it would require more demonstrations to have sufficient training data coverage, which can make it less data efficient.

Figure 5 shows real-world policy rollout results. We notice different failure modes for baseline methods. For diffusion policy and diffusion policy with RGBD, when a novel instance is presented, they may stop early and fail to make progress. Due to their reliance on 2D observations, they cannot generalize to unseen instances with varying appearances and geometries. Ours *without* semantics often fail to differentiate geometric ambiguity, like knife handles, while our method could identify the right part for manipulation. Our method also focuses on geometric details, like the soda can tab to accomplish the can flipping task, while ours *without* semantics places the soda can upside down.

### 4.3 Generalization to Novel Configurations and Instances

We also analyze how well our method can generalize to novel configurations and instances. Figure 6 illustrates the predicted actions of different policies under the same observations. For the seen instances, as shown in the leftmost column, we can see that the predicted trajectories for all policies in the first example point towards the knife handle correctly. However, when the knife direction changes, our method *without* semantics fails to respond accordingly, while the diffusion policy and our method predict actions accordingly. This demonstrates that our method can understand the semantic difference between knife handle and knife blade, which results in safe robot behaviors.

When the novel instance is presented with similar configurations, our method will predict a successful trajectory guiding toward the knife handle, while diffusion policy will predict a nonsmooth trajectory, which shows that it is brittle to environmental factors and lacks category-level generalization capabilities. Although our method *without* semantics predicts a smooth trajectory, it approaches the knife blade, which is unsafe.

### 4.4 Generalization Analysis

We visualize the 3D semantic fields in Figure 7 by overlaying them with the raw point cloud. These semantic fields benefit the policy for two primary reasons: 1) Highlighted points are semantically meaningful and crucial for task completion. For example, to collect books on the shelf upright, the

robot must recognize the book titles. Geometric information alone is inadequate for task completion. 2) Semantic fields are consistent across different instances. The mug example shows that the activation on the handle is consistent across mugs with various appearances and poses. The category-level consistency enables our method to achieve category-level generalization. We also present t-SNE analysis in the supplementary material.

## 4.5 Extension to Other IL Methods

We also extended our 3D semantic fields and showed benefits to other imitation learning (IL) methods, such as Action Chunking with Transformer (ACT) [65]. The detailed results are presented in the supplementary material.

## 5 Conclusion

In this work, we proposed a diffusion policy framework leveraging 3D semantic representations, with policies modeled as DDPMs conditioned on the semantic fields alongside point clouds. We conduct comprehensive physical experiments to show that our framework can generalize to novel instances within the category, resolve geometric ambiguity, and resolve nuanced geometric details. Compared with the vanilla diffusion policy with 20% success rate, our method can reach 93% success rate on unseen instances, demonstrating the effectiveness of our method.

**Limitation.** For each task, we select a set of fixed reference features to construct semantic features. However, for long-horizon and fine-grained tasks, such as making coffee and assembling, the policy's attention needs to adapt as tasks progress. Constructing task-specific

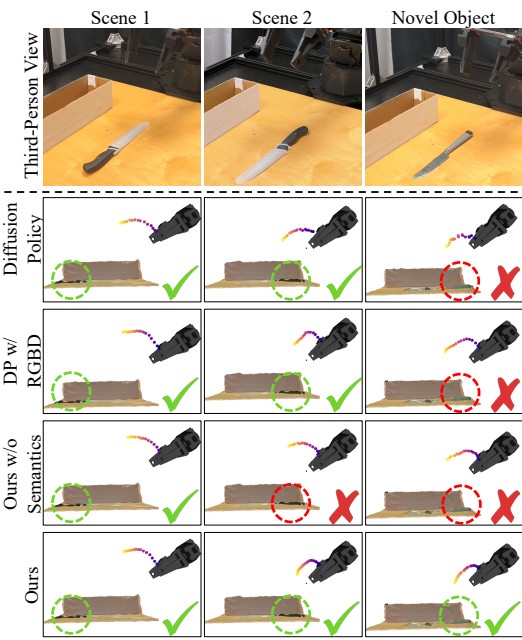

Figure 6: **Predicted Action Comparisons.** We compare different policies' responses under the same state. Diffusion policy's predicted actions change with knife direction accordingly but are not smooth for novel instances. Due to geometric ambiguity, our method *without* semantics fails to distinguish knife directions, and predicts similar actions even under different knife directions, while our method consistently recognizes the knife handle position and predicts correct actions.

semantic fields that can adaptively select the abstraction level, can be a future direction. In addition, our method could become more interpretable and potentially more efficient if we can incorporate the geometric properties in a more explicit fashion.

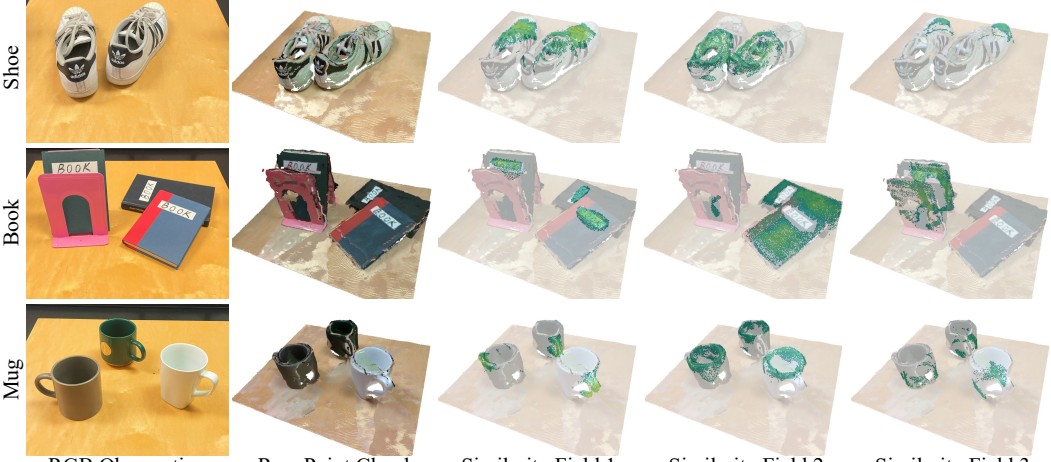

Figure 7: **3D Semantic Fields Visualization.** We visualize 3D semantic fields in the scene with multiple instances from the same category. 3D semantic fields exhibit similar patterns across different instances, such as highlighted book titles of all books. Furthermore, these highlighted areas represent semantically meaningful parts and are important for various tasks, such as shoelaces, shoe heads, book stands, mug handles, and so on.

## Acknowledgement

This work is partially supported by Sony Group Corporation. The opinions and conclusions expressed in this article are solely those of the authors and do not reflect those of any Sony entity.

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
