# OpenReview forum: "GenDP: 3D Semantic Fields for Category-Level Generalizable Diffusion Policy"
_robot-learning.org/CoRL/2024/Conference — CoRL 2024_

### Official Review · Reviewer_ruvW · 2024-07-20
**Combination of D3fields and diffusion policy, impactful contribution, extensive analysis**

**Originality:** 3
**Technical Quality:** 4
**Clarity Of Presentation:** 4
**Potential Impact:** 4
**Recommendation:** 3
**Confidence:** 5

**Review:**

The authors propose a generalizable 3D diffusion policy trained on the complete point cloud of the scene in conjunction with a high-dimensional descriptor field. The descriptor field processes segmentation and feature extraction via foundation model DINOv2 to construct a 3D feature cloud from RGBD images (4 cameras). The constructed feature cloud is merged with the scene point cloud for input into diffusion policy. Extensive comparison is done for generalization in terms of instance, geomtery and category. Real world experiments are also shown for the ALOHA setup.

Strengths:
This paper is built upon D^3 fields to extract the feature clouds with diffusion policy. The resulting GenDP is able to leverage semantic information to perform better than diffusion policy with raw point cloud information only.

Exploiting the representations captured by DINOv2, the trained policy is able to understand different parts of the object and correlate between seen and unseen objects of same category. for example, handles of knife and heads of different brush.

Weaknesses:
The work has limited novelty (considering diffusion policy with 3D point cloud input well explored [2]), relies on the complete point cloud of the scene and the object (more specifically) and is limited to short task lengths.

I thank the authors for presenting their method clearly. I consider this paper as an important contribution towards generalizable manipulation.

[1] https://arxiv.org/abs/2309.16118/
[2] https://arxiv.org/abs/2403.03954

**Quality Of The Limitations Section:**

3

**Questions For Rebuttal:**

1. Can you provide reference images and features for tasks other than shoe? Why do you need reference features for your problem statement? What happens if you only merge scene point cloud and DINOv2 feature cloud?

2. Does the performance change if we use different reference (images I suppose from D3fields)? How do you choose them?

3. Can you clarify figure 7 further? How are you generating different similarity fields for different parts of the object?

The above clarifications can help in understanding the capabilities of the proposed approach.

**Robotics Focus:**

4

**Summary Of Paper:**

Generalizable Diffusion policy with information about the complete 3D point cloud and semantics

**Summary Of Recommendation:**

The proposed method is a strong advancement towards generalizable manipulation. However further clarifications are required.

---

### Official Review · Reviewer_WbYD · 2024-07-21
**No main novel contribution, just a different input representation**

**Originality:** 1
**Technical Quality:** 2
**Clarity Of Presentation:** 4
**Potential Impact:** 2
**Recommendation:** 1
**Confidence:** 4

**Review:**

The authors propose a pipeline for incorporating semantic information into pointclouds reconstructed from multiple views via descriptors/embeddings from a Vision Foundation Model.  This "3D Semantic Fields"  is then used for downstream imitation learning with a PointNet++ encoder.

The main focus of the paper is the use of 3D semantic fields i.e. Pointcloud with dense feature descriptors with Diffusion Policy (DP). This itself doesn't warrant a novel contribution as it neither proposes any novelty in the feature descriptor input or the learning, nor is there any specific improvement done on top of DPs. It's just the use of a semantic pointcloud representation with a specific encoder suitable for such an input. While the approach shows improvement over vanilla DPs or vanilla DP with  RGBD input, or the approach without semantic pointclouds but just RGBD pointclouds, there could have been some fairer comparison with other works or with different abalations to show the success of the approach.

Major Comments:
- L31-L32 "we augment geometric information with semantics" -> Many papers that already do this
- "resolve geometric ambiguity" It is unclear what is the main way this is resolved other than using the semantic information for resolving instead of the 3D aspects.
- "and attend to subtle geometric details" More an outcome of the learning, than an explicit methodological/theoretical contribution. Additionally, rather than attending to the geometric details, it is the semantics that gets attended on that help resolve ambiguities, as mentioned in L188-189 " First, they help to attend to geometric details, such as the can tab, to place the can on the table upright, while ours without semantics cannot when solely relying on geometry information" This shows that it does not rely on the geometry, but more on the semantics that causes the main increase in performance.
- Line 35-36 "we introduce a novel diffusion policy framework" This is an overstatement since they just use a different encoder with off-the-shelf Diffusion policy imitation learning. There is no novelty as such in the diffusion aspect nor the policy learning, nor the representation learning.
- Line 36 "uses a scene representation in the form of 3D semantic fields." This is also done by many works, some of which are also mentioned in the related works. In L103, The authors mention that "In contrast, we extract 3D descriptor fields without distillation", however, it's not explained why that is a better approach, since some similar approaches work well with such neural feature fields for extracting semantically similar areas for doing robot manipulation.

  Florence, Peter R., Lucas Manuelli, and Russ Tedrake. "Dense Object Nets: Learning Dense Visual Object Descriptors By and For Robotic Manipulation." Conference on Robot Learning. PMLR, 2018.

  Xu, Zhenjia, et al. "Learning 3D Dynamic Scene Representations for Robot Manipulation." Conference on Robot Learning. PMLR, 2021.

  Shen, William, et al. "Distilled Feature Fields Enable Few-Shot Language-Guided Manipulation." Conference on Robot Learning. PMLR, 2023.

- L52: "to disambiguate vague geometric information" It's not that the geometric info is disambiguated, it's just that the semantic information enables the detection of relevant features irrespective of the geometry (as seen in the ablation in the appendix), which is the general contribution of such 3D semantic pointclouds

  Gu, Qiao, et al. "Conceptgraphs: Open-vocabulary 3d scene graphs for perception and planning." arXiv preprint arXiv:2309.16650 (2023).

- L127-L128 "We fuse features from multiple viewpoints by applying a weighted sum, thus obtaining the descriptor f corresponding to p" This is  similar to doing a distilliation, which the authors claim in L103 is not what they do. Here are some similar works for doing such feature fields or dense 3D descriptor pointclouds.

  Zhou, Shijie, et al. "Feature 3DGS: Supercharging 3D Gaussian Splatting to Enable Distilled Feature Fields." arXiv preprint arXiv:2312.03203 (2023).

  Shen, William, et al. "Distilled Feature Fields Enable Few-Shot Language-Guided Manipulation." Conference on Robot Learning. PMLR, 2023.

- While comparing DP+RGBD, rather than using a standard DP pipeline, using a more RGB-D oriented encoder, like Red-Net, or RD3D would have been a fairer comparison since a pointcloud specific encoder is one of the main contribution of the work. Additionally, using a Semantic-informed encoder on the RGB with diffusion policy would be another tangible baseline to run for a fairer evaluation of a semantically-aware Diffusion policy. Moreover, the current RGBD pointcloud comes from multiple views whereas a standard DP would use just a single input. Using a single view pointcloud, would be a fairer comparison to better understand how the partial observability plays a role in the learning.


It is due to these shortcomings and more importantly the lack of a strong novel contribution that makes me unable to recommend the paper for acceptance.

**Quality Of The Limitations Section:**

1

**Questions For Rebuttal:**

1. What is the main novelty or contribution of the paper? How is the 3D semantic fields you show different from other works in literature that augment pointclouds with additional descriptors?
2. Please elaborate on how the approach disambiguates geometric information (which is not that it uses semantic information)
3. Please comment on the issue of baselines as mentioned in the review
4. In the explanation of attending to subtle geometric details, you highlight that the semantics helps in understanding where to grasp, however, this does not explain how the "geometric details" help the process, but rather it's the semantics that helps the learning.

**Robotics Focus:**

4

**Summary Of Paper:**

Imitation Learning via Diffusion Policy using a multi-view semantically segmented pointcloid ("3D semantic field") as input instead of single view image inputs

**Summary Of Recommendation:**

No clear novelty, just the use of a different input representation to a standard Diffusion Policy Imitation Learning.

---

### Official Review · Reviewer_m8cy · 2024-07-24
**Using Semantic Fields for Generalizable Robotics Manipulation**

**Originality:** 3
**Technical Quality:** 3
**Clarity Of Presentation:** 5
**Potential Impact:** 3
**Recommendation:** 3
**Confidence:** 4

**Review:**

The paper is easy to read and the motivation is clear. Despite the impressive results of Diffusion Policies, these models lack proper generalization capabilities, in part due to the lack of proper representation of the perception. This work explores the option of using 3D semantic fields as perception representation for improving the generalization capabilities of diffusion policies.

**Strenghts**
- The selected problem is sound and relevant for the robotics community. Figuring out strategies for improved generalization is a highly relevant topic for the community.
- The selected experiments are good. Through the evaluation phase, they evaluate the influence of the semantic fields and 3D representations for generalization. The evaluation is well aligned with the contribution they are aiming to have with their presented work.

**Weaknesses**
- The authors fail to properly explain how the semantic fields are built. In the experimental section, they claim they collected 100 demonstrations for the insert pencil task. For computing the semantic fields, they make use of reference descriptors. How do you obtain them? Which images do you use from the 100 demonstrations? The authors could spend a paragraph explaining exactly how these reference descriptors are obtained.
- A highly related work is '3D Diffuser Actor: Policy Diffusion with 3D Scene Representations'. The author cites the work but they fail in explaining what makes their work different from theirs. Given both methods use 3D feature fields and diffusion policy, it might be interesting to add some lines highlighting the benefit of their method wrt. this one.
- Despite the interesting properties of the semantic fields, the authors fail to exploit properly their geometric properties. In their work, they take PointNet++ and transform the 3D semantic field into a visual feature vector that is fed into the diffusion policy. This differs from other works using Semantic Fields, such as Distilled Feature Fields that exploit the 3D geometry of the perception to generate actions. The authors could highlight this as a limitation and propose approaches for extension.
- Both semantic fields and diffusion policies are not novel in the community and the authors do not propose any geometrically meaningful approach to exploit the semantic field, but simply combine the parts through a visual feature representation.

**Quality Of The Limitations Section:**

2

**Questions For Rebuttal:**

- How do you select the images for the reference descriptors? Is it automatized? Could the reference descriptors represent background elements that are not relevant for solving the task?

**Robotics Focus:**

4

**Summary Of Paper:**

The authors aim to solve a behavioral cloning problem with diffusion policies. To make the model for generalizable, they adapt the perception pipeline with a 3D semantic field. They show that adding the following increases the generalization capabilities wrt. the baselines.

**Summary Of Recommendation:**

The paper introduces an interesting approach to improve the generalization of diffusion policies, by using 3D semantic fields as perception. The work is interesting. Nevertheless, both sematic fields and diffusion policies are not novel in the community and the authors do not propose any geometrically meaningful approach to exploit the semantic fields.Thus, it is a weak accept for me.

---

### Author Rebuttal · Authors · 2024-08-13

Our revised main paper and supplement are in the rebuttal file.

---

### Decision · Program_Chairs · 2024-09-04

**Decision:**

Accept

**Comment:**

Strengths:
+ Paper is written clearly.
+ GenDP achieves compelling results with unseen objects.
+ Results include ablations and evaluations that showcase the generalization aspects of GenDP.

Weaknesses:
- The approach is not entirely novel. Several prior manipulation works use 3D semantic fields as noted by all the reviewers.
- Details on how reference descriptors are obtained is missing.
- Diffusion Policy baseline should use multi-view input instead of single-view RGB-D for a fairer comparison.
- GenDP needs to be better positioned among existing works like 3D Diffuser Actor, 3D Diffusion Policies, etc.

Post rebuttal:
The author responses clarified some of the concerns regarding novelty. Additional experiments ablated different encoders.